# New Design Scheme for and Application of Fresnel Lens for Broadband Photonics Terahertz Communication

**DOI:** 10.3390/s24237592

**Published:** 2024-11-27

**Authors:** Peng Tian, Yang Han, Weiping Li, Xiongwei Yang, Mingxu Wang, Jianjun Yu

**Affiliations:** State Key Laboratory of ASIC and System, Key Laboratory for Information Science of Electromagnetic Waves (MoE), School of Information Science and Technology, Fudan University, Shanghai 200433, China; 22210720225@m.fudan.edu.cn (P.T.); 22210720136@m.fudan.edu.cn (Y.H.); 19210720141@fudan.edu.cn (W.L.); 22110720137@m.fudan.edu.cn (X.Y.); 21110720068@m.fudan.edu.cn (M.W.)

**Keywords:** terahertz wireless transmission, plano-convex lens, Fresnel lens, PTFE

## Abstract

In terahertz communication systems, lens antennas used in transceivers are basically plano-convex dielectric lenses. The size of a plano-convex lens increases as the aperture increases, and thinner lenses have longer focal lengths. Through theory and simulation, we designed a Fresnel lens suitable for the terahertz band to meet the requirements of large aperture and short focal length, and simulated the performance, advantages, and disadvantages of the terahertz Fresnel lens. A 300 GHz terahertz wireless communication system was built to verify the gain effect of the Fresnel lens antenna. The experimental results demonstrate that the Fresnel lens can be used for long-distance terahertz communication with larger aperture diameters, overcoming the limitations of traditional plano-convex lenses. The theoretical gain of a 30 cm Fresnel lens is 48.83 dB, while the actual measured gain is approximately 45 dB.

## 1. Introduction

Existing low-frequency spectrum resources have become saturated, and higher-frequency terahertz waves undoubtedly have great application prospects. The frequency of terahertz waves is 0.1–10 THz, and the band is between microwaves and far-infrared light. It can be considered that terahertz waves are in the transition zone from macroscopic classical theory to microscopic quantum theory. Since the wavelengths of terahertz waves are close to the visible light region and their penetrating ability is stronger than that of visible light, optical methods can be used to design dielectric lenses with reasonable structure and size to focus terahertz waves [1,2,3,4].

In current research on terahertz transmission systems, the dielectric lenses widely used on transmitters and receivers are typically conventional plano-convex lenses [5,6,7,8,9,10,11,12]. The design of plano-convex lenses is generally based on the ray-tracing method in geometric optics, where the feed source is typically positioned to coincide with the focal point. In this design approach, the curvature of the lens is usually determined using Fermat’s principle and Snell’s law. Plano-convex lenses have become widely adopted optical components in terahertz transmission systems due to their stable focusing capability, wide field of view, and good multi-frequency applicability. In practice, even when there is some angular deviation between the direction of the received signal and the lens surface, plano-convex lenses can still maintain a high gain performance. Additionally, the design of plano-convex lenses is straightforward, and the manufacturing technology is well-established, with design schemes based on geometric optics being particularly mature. Consequently, plano-convex lenses are extensively used in current research on terahertz transmission systems to enhance antenna gain and directivity, thereby facilitating long-distance signal transmission [13,14,15,16,17,18,19,20,21].

In the literature [7], a research team from Fudan University established a long-distance terahertz signal transmission system that achieved a transmission distance of 850 m. During the experiment, the researchers installed a plano-convex lens with a diameter of 10 cm in front of the transmitting antenna to correct phase and collimate the electromagnetic beam, while another plano-convex lens with a diameter of 30 cm was placed in front of the receiving antenna to focus the electromagnetic beam. The experimental results indicated that the gain of the antennas reached 48.5 dB and 65 dB, respectively. Furthermore, other studies have also demonstrated the critical role of polytetrafluoroethylene (PTFE) lenses in long-distance terahertz wireless transmission, with the lens gains at both the transmitter and receiver reaching 70 dB [10,22]. These findings underscore the indispensable role of high-gain dielectric lenses in long-distance terahertz transmission systems.

As the transmission distance increases, terahertz signals experience greater atmospheric losses and free-space path losses during propagation, necessitating higher gain for lens antennas. Typically, assuming structural losses are negligible, doubling the effective aperture of a plano-convex lens results in an approximate 6 dB increase in antenna gain (this conclusion can be directly derived from Formula (12)). Therefore, in long-distance terahertz transmission, increasing the aperture of the lens antenna becomes a key method for enhancing antenna gain. However, in practice, as the aperture size increases, the volume and weight of the plano-convex lens also increase significantly. Additionally, due to the inverse relationship between the lens focal length and thickness, achieving a balance with short-focal-length large-aperture lenses in engineering practice presents considerable challenges [13,14,15].

To address the aforementioned challenges, we propose a novel terahertz lens antenna design—the Fresnel lens. The Fresnel lens consists of a series of concentric prisms etched onto the lens’ base surface, making it lighter and thinner than traditional plano-convex lenses. This characteristic allows the Fresnel lens to maintain a reduced thickness and mass even in scenarios involving large apertures and short focal lengths, thereby helping to minimize material absorption losses and facilitating easier engineering deployment and installation [23,24].

The Fresnel lens, as an important optical component, has been widely applied across various fields. Owing to its unique structural design, the Fresnel lens can significantly reduce both the thickness and weight of the lens while maintaining optical performance, making it particularly advantageous for use in portable devices. For instance, in solar concentrators, Fresnel lenses focus parallel light onto a small area, enhancing energy capture efficiency. In lighting systems, Fresnel lenses are extensively used in lamps and automotive headlights to achieve efficient optical designs. Additionally, in photography and projection, Fresnel lenses are employed as light-focusing and -diffusing devices due to their lightweight nature and cost-effectiveness. These diverse applications underscore the indispensable role of Fresnel lenses in modern optical systems, highlighting their significant academic and engineering value [25,26,27,28,29]. There has been some research on Fresnel lenses in the terahertz field. Reference [30] used scalar diffraction theory to study the spatiotemporal and spectral characteristics of ultra-wideband terahertz pulses after passing through a Fresnel lens, and proposed the idea of using a Fresnel lens as a terahertz filter. Reference [31] and some commercial companies have manufactured small-aperture terahertz Fresnel lenses, but there has been no practice of large-aperture Fresnel lenses in actual terahertz wireless systems.

In this study, we have designed a Fresnel lens operating in the terahertz frequency band and successfully validated its performance within a terahertz wireless transmission system. We employed geometric optics methods to design the Fresnel lens, precisely calculating its parameters based on Snell’s law, and conducted thorough ray-tracing analyses using optical software (Zemax OpticStudio 19.4). Through comparative simulations, we found that, although the electromagnetic beam-focusing ability of the Fresnel lens is slightly lower than that of traditional plano-convex lenses, it offers a shorter focal length and lighter weight, making it particularly advantageous in large-aperture applications. In practical validation within a terahertz wireless transmission system, the Fresnel lens demonstrated significant potential. This design offers a promising research direction for the development of high-gain terahertz lens antennas in the future.

The remainder of this paper is organized as follows: Section 2 provides a brief overview of the relevant theory of Fresnel lenses, followed by design analysis and simulation. Section 3 discusses the performance and actual gain of the fabricated Fresnel lens within a terahertz wireless communication system. Section 4 presents the results and conclusions of this work.

## 2. Theory and Simulation

### 2.1. Approximate Treatment of Geometric Optics

As mentioned earlier, the terahertz frequency band is close to the visible light region, allowing it to be approximated using geometric optics [32]. The following provides a proof of this approximation. Consider a general time-harmonic field in an isotropic non-conductive medium, as follows:(1)$$\begin{array}{l} E(r,t)=E_{0}(r)e^{-i\omega t}\\ H(r,t)=H_{0}(r)e^{-i\omega t} \end{array}$$

The refractive index of the medium is $n=\sqrt{\varepsilon \mu }$, ε is the relative dielectric constant, and μ is the relative magnetic permeability. $E_{0}$ and $H_{0}$ represent the complex vector function of the position. In the passive region, without considering the time factor, the above vectors satisfy Maxwell’s equations, as follows:(2)$$\begin{array}{c} \nabla \times H_{0}+ik_{0}\varepsilon E_{0}=0\\ \nabla \times E_{0}-ik_{0}\mu H_{0}=0\\ \nabla \cdot \varepsilon E_{0}=0\\ \nabla \cdot \mu H_{0}=0 \end{array}$$
where $k_{0}=\omega /c=2\pi /\lambda_{0}$, $\lambda_{0}$ is the vacuum wavelength. Consider the following:(3)$$\begin{array}{l} E_{0}(r)=E_{0}{(r)e}^{ik_{0}S(r)}\\ H_{0}(r)=H_{0}{(r)e}^{ik_{0}S(r)} \end{array}$$

Evidently, $E_{0}$ and $H_{0}$ are vector functions of position, representing the amplitude. S is a real scalar function of position, representing the phase function of the propagating wave.
(4)$$\begin{array}{l} \nabla \times H_{0}=(\nabla \times H_{0}+ik_{0}\nabla S\times H_{0})e^{ik_{0}S}\\ \nabla \cdot \mu H_{0}=(\mu \nabla \cdot H_{0}+H_{0}\cdot \nabla \mu +ik_{0}\mu H_{0}\cdot \nabla S)e^{ik_{0}S} \end{array}$$

Similarly, by substituting $\nabla \times E_{0}$ and $\nabla \cdot \varepsilon E_{0}$ into the Maxwell’s equations, we obtain the following:(5)$$\begin{array}{c} \nabla S\times H_{0}+\varepsilon E_{0}=-\frac{1}{ik_{0}}\nabla \times {\;H}_{0}\\ \nabla S\times E_{0}-\mu H_{0}=-\frac{1}{ik_{0}}\nabla \times {\;E}_{0}\\ E_{0}\cdot \nabla S=-\frac{1}{ik_{0}}(E_{0}\cdot \nabla \ln\varepsilon +\nabla \cdot E_{0})\\ H_{0}\cdot \nabla S=-\frac{1}{ik_{0}}(H_{0}\cdot \nabla \ln\mu +\nabla \cdot H_{0}) \end{array}$$

Let us consider the magnitude of $k_{0}$. In the terahertz frequency range, as well as in the visible light spectrum, the electromagnetic field is characterized by relatively fast oscillations (with frequencies in the order of 10^12^Hz and above), or in other words, relatively short wavelengths (in the order of 10^−3^ cm and below). Therefore, it can be expected that, under such circumstances, ignoring the finite size of the wavelength can provide a good first-order approximation of the wave propagation laws. In traditional optical research, it is common practice to ignore the wavelength, i.e., $\lambda_{0}\to 0$. This branch of study is typically referred to as geometric optics because, under this approximation, the laws of wave propagation can be described using geometric language. In this case, the energy of the electromagnetic field can be considered to be transmitted along certain curves (rays). For issues related to the terahertz frequency range, such an approach remains suitable. Additionally, since the size of the lens antenna we designed is much larger than the wavelength in free space, diffraction effects can be ignored to some extent. Therefore, it is effective to use the assumptions of geometric optics to design these lens antennas. Thus, when $\lambda_{0}\to 0$, $k_{0}\to +\infty$, and we can omit all product terms on the right side of the equation.
(6)$$\begin{array}{c} \nabla S\times H_{0}+\varepsilon E_{0}=0\\ \nabla S\times E_{0}-\mu H_{0}=0\\ E_{0}\cdot \nabla S=0\\ H_{0}\cdot \nabla S=0 \end{array}$$

Evidently, by taking the dot product of ∇S with the first two equations, the last two equations can be derived. Therefore, by directly combining the first two equations and substituting the refractive index, we obtain the following:(7)$$\nabla S\times (\nabla S\times E_{0})+n^{2}E_{0}=0$$

Further transformation yields the following:(8)$$|\nabla S{|}^{2}=n^{2}$$

This is the fundamental equation of geometric optics: the eikonal equation, where S is called the eikonal, n is the refractive index in space, and ∇S represents the phase gradient (i.e., the direction of the light rays). In an isotropic homogeneous medium, the refractive index does not vary with position, so S is constant.

The eikonal equation indicates that, under the geometric optics approximation, the propagation path of the light rays is determined by the gradient of the phase function S(r). Specifically, light rays always travel in the direction where the phase changes most rapidly, which is consistent with Fermat’s principle: light always chooses the path with the shortest optical length when traveling between two points. From Fermat’s principle, the fundamental laws of reflection and refraction in geometric optics can be derived [32,33,34].

### 2.2. The Principle of the Fresnel Lens

The Fresnel lens is an evolution based on the plano-convex lens (or aspheric lens). In traditional geometric optics, light rays are refracted at the surface where different media meet, altering their direction of propagation. Therefore, the contour of the refractive surface of a traditional lens largely determines its focusing ability, while the material between the refractive surfaces does not change the direction of light propagation but instead significantly increases material absorption losses and weight [35,36,37]. Essentially, the Fresnel lens removes the portions of the original lens that do not affect the curvature of the lens, retaining only the parts that effectively refract light, as illustrated in Figure 1.

In engineering practice, considering the challenges of design and manufacturing, the annular surfaces of a Fresnel lens are generally not curved but conical. The base of the lens is composed of a series of concentric rings, and from a cross-sectional view, each ring can be considered a small prism with a different tilt angle. Therefore, this design method is also known as the prism method [37,38,39]. The terahertz waves refracted by the ring surfaces are focused on the focal point, and the focus position of the lens can be adjusted by changing the tilt angle of the rings. As shown in Figure 1d, the refraction of a single annular ring is considered. Let h be the height of the incident light, $\theta_{BL}$ the angle of the lens ring, $\theta_{f}$ the angle between the refracted light and the optical axis, and F the focal length of the lens. According to the law of refraction, the following relationship can be derived:(9)$$\begin{array}{l} n\sin\theta_{BL}=\sin(\theta_{BL}+\theta_{f})\\ \tan\theta_{f}=\frac{h}{F} \end{array}$$

The angle of each ring and its distance from the center of the lens can be calculated through iterative optimization. In this study, mature optical design software (Zemax OpticStudio 19.4) is used for the design process, allowing for weighted optimization of multiple parameters.

### 2.3. The Structural Loss and Frequency Dependence of the Fresnel Lens

As shown in Figure 2a, for a Fresnel lens, when light rays are incident on the surface of the lens rings, they are refracted, but at the edges of each ring, some rays are blocked and cannot reach the designated focal point, resulting in structural loss. Let the tooth width of the lens ring be w, and within the annular region of width h for that ring, the incident light rays are obstructed by the side surfaces (non-working surfaces) of the lens ring, preventing them from converging on the focal plane. Instead, they scatter, forming off-axis rays, as illustrated in Figure 2b. At this point, the structural loss rate for the ring is η=h/w. Clearly, this type of structural loss is unavoidable. Generally, the wider the lens ring section (i.e., the larger w), the weaker the scattering effect.
(10)$$n^{2}(\lambda )=1+\sum_{i}\frac{B_{i}\lambda^{2}}{\lambda^{2}-C_{i}}$$

The above equation is the Sellmeier dispersion formula, where $B_{i}$ and $C_{i}$ are material-dependent constants, λ is the wavelength, and n is the refractive index. From Equation (9), it is clear that the refractive index is a crucial parameter in lens structure design. According to the Sellmeier equation, in the same medium, the refractive index varies with different wavelengths of light. The refractive index typically decreases as the wavelength increases, reflecting the dispersion characteristics of the material—meaning that light of different wavelengths travels at different speeds in the material, leading to dispersion effects. Another significant impact of the dispersion effect on the Fresnel lens is its frequency sensitivity. The refractive surface of a plano-convex lens is a smooth curve, and its design is considered as a whole. However, the Fresnel lens is designed through sectional optimization, meaning that the parameters of the lens rings are determined based on the design wavelength. For slightly different wavelengths, these parameters become inaccurate. Generally, the bandwidth of the lens is inversely related to the number of rings.

### 2.4. Design and Simulation

Before designing, it is essential to determine the material for lens processing. Polytetrafluoroethylene (PTFE) is commonly used as a lens material in the sub-terahertz and terahertz frequency bands [40]. It is a white, hard, and heavy plastic. Its density is around 2.2 g/cm^3^. PTFE melts at 327 °C and retains its useful properties in a wide temperature range between −73 °C and 204 °C [41,42]. Its refractive index is 1.4 in a wide wavelength range. PTFE has a low dielectric constant, about 1.96 at 520 GHz, which means its insertion loss is relatively low [43]. See Figure 3.

We designed a Fresnel lens with an effective aperture of 30 cm based on a frequency of 300 GHz. The parameters of the Fresnel lens were iteratively adjusted using the prism method. Afterward, the designed model was subjected to 3D modeling and ray tracing in optical software (Zemax OpticStudio 19.4). The results are shown in Figure 4. The higher sidelobes of the Fresnel lens are due to structural losses. During the optimization process, we appropriately reduced the number of rings on the Fresnel lens, considering the limitations of practical manufacturing.

It is worth noting that, compared to plano-convex lenses, Fresnel lenses are highly sensitive to changes in the incident angle. We now simulate the energy distribution on the focal plane for lens receiving angles of 0°, 3.5°, and 5°, respectively. As shown in Figure 5, as the incident angle of the Fresnel lens increases, the on-axis radiation drops dramatically. This situation is also caused by the sectional design of the Fresnel lens, which is a structural defect that can be mitigated by wave collimation. In comparison, changes in the incident angle have a much smaller impact on plano-convex lenses. Therefore, plane equidistant Fresnel lenses are extremely sensitive to changes in the incident light angle, which imposes higher requirements on terahertz antenna alignment technology.

### 2.5. Estimation of Lens Antenna Gain

A lens antenna is considered an aperture antenna, and its gain can be estimated using the following formula, based on the effective aperture:(11)$$G=\frac{4\pi A_{e}}{\lambda^{2}}$$

$A_{e}$ is the effective area of the antenna and λ is the signal wavelength. Taking into account the antenna efficiency η and the antenna diameter D (effective aperture), we obtain the following:(12)$$G_{dB}=10\log_{10}\left(\eta {\left(\frac{\pi D}{\lambda }\right)}^{2}\right)$$

For traditional lens antennas, antenna efficiency refers to the ability of the antenna to effectively convert input power into radiated power. Antenna efficiency is typically influenced by several factors, such as material absorption losses, surface roughness, matching losses, reflection losses, environmental factors (temperature and humidity), edge diffraction, and the lens’s focal length (F-number). In the case of Fresnel lenses, special attention must also be paid to structural losses and frequency dependence. Additionally, the sensitivity to incident angles may increase matching losses.

Given these complexities, accurately calculating the gain of a lens antenna can be challenging. Instead, we can use the Strehl ratio calculated by optical software as a simplified substitute for antenna efficiency in our estimation. The Strehl ratio is defined as the ratio of the peak intensity on the optical axis in the actual beam’s far-field to that of an ideal beam with the same power and uniform phase. It is a measure of the quality of the optical energy distribution in an optical system.

We performed optical simulations for both a plano-convex lens and a Fresnel lens with an effective diameter of 30 cm, resulting in the relative irradiance intensity curves on the focal plane for the two lenses, as shown in Figure 6. The focal length of the Fresnel lens is 35 cm and the maximum thickness is 15 mm. The focal length of the plano-convex lens is 50 cm and the maximum thickness is 55 mm. Using the Strehl ratio as the peak value and with a working wavelength of 1 mm (300 GHz), the calculated gains for the plano-convex lens and the Fresnel lens are 55.44 dB and 48.83 dB, respectively, with a difference of approximately 6.61 dB. It is important to note that the Strehl ratio is an indicator of image quality, and, when there is significant aberration, the Strehl ratio can be very low. Therefore, the estimated gain using this method is likely lower than the actual gain.

## 3. Experimental Setup

We designed a Fresnel lens with an effective diameter of 30 cm, based on the 300 GHz frequency band, and fabricated it using PTFE. After designing and planning the relevant parameters of the Fresnel lens, we generated the corresponding 3D model and manufactured it using a computer numerical control (CNC) machine. To validate the performance of the Fresnel lens, we set up a terahertz wireless transmission system.

Figure 7 shows the experimental setup for photon-assisted 300 GHz terahertz signal transmission over a 50 m wireless link. The baseband signal generation process is carried out offline in MATLAB R2022a at the transmitter. Initially, the bit sequence is mapped into QAM (quadrature amplitude modulation) signals, followed by digital-to-analog conversion (DAC). This digital signal is then loaded into a 120 GSa/s arbitrary waveform generator (AWG) to generate QAM signals at different baud rates. These signals are then amplified by two parallel electrical amplifiers (EAs) with 25 dB gain each. The setup incorporates two free-running external cavity lasers (ECLs), labeled as ECL-1 and ECL-2, each with a linewidth of 100 kHz. ECL-1 emits a continuous optical wave at 1550.00 nm, while ECL-2 operates at 1552.4 nm. The I/Q modulator, which has a 30 GHz 3 dB bandwidth, is driven by an optical carrier emitted by ECL-1, completing the optical modulation process. The signal, amplified to post-modulation by a polarization-maintaining erbium-doped fiber amplifier (PM-EDFA), is then combined with another optical carrier emitted by ECL-2 using a polarization-maintaining optical coupler (PM-OC). The transmitted signal, after traversing 100 m of SMF-28 (Single-Mode Fiber-28), is regulated in optical power using a variable optical attenuator (VOA) before entering the uni-traveling-carrier photodiode (UTC-PD). Inside the UTC-PD, 300 GHz THz signals are generated through optical heterodyne technology. The amplified signal is then transmitted into free space through a transmitter horn antenna (HA-1) with a 25 dB gain. We placed a pair of lenses, Lens-1 and Lens-2, in front of the transmitter and receiver antennas. Lens-1 is a 10 cm diameter plano-convex lens, and Lens-2 is the 30 cm diameter Fresnel lens designed for this experiment. Due to the Fresnel lens’ sensitivity to incident angles, the precise alignment of Lens-1 and Lens-2 is crucial. Inadequate alignment can result in significant signal loss, negatively impacting the overall system performance. We used some auxiliary tools for correction. Our transmitter and receiver are set up in the corridors of the laboratory building at the Jiangwan campus of Fudan University, where the indoor temperature and humidity are well controlled.

At the wireless receiver end, the signal initially undergoes amplification by a low-noise amplifier (LNA), which provides a gain of 33 dB. Subsequently, a local oscillator (LO) signal undergoes 12-fold amplification in the mixer. It is then mixed with the THz signal and down-converted to an intermediate frequency (IF) signal. After down-conversion, the IF signals are further amplified by EA-3 with a gain of 26 dB. Finally, these signals are captured by a digital storage oscilloscope (OSC). The parameters of key devices in the experimental setup are shown in Table 1. Figure 7d shows photos of the transmitter and receiver setups.

The obtained IF signal, captured by the OSC, undergoes offline digital signal processing (DSP) in MATLAB R2022a. The offline DSP process at the receiver (Rx) involves multiple stages: down-conversion, low-pass filter, Gram–Schmidt orthogonalization procedure (GSOP), resampling, T/2 constant modulus algorithm (CMA) equalization, frequency offset estimation (FOE), blind phase search (BPS), and direct detection minimum mean square error (DD-LMS) equalization. Initially, the intermediate frequency signal is shifted to baseband through down-conversion. Next, the baseband signal passes through a low-pass filter to remove high-frequency noise, and the GSOP is applied to address IQ imbalance. Subsequently, CMA linear equalization and DD-LMS compensation address the inter-symbol interference caused by channel propagation characteristics. The FOE compensates for the frequency drift effect induced by laser frequency offset. The BPS improves the phase noise introduced by the linewidth of the laser. The distinct procedures pertaining to the transmitter DSP (Tx-DSP) and the receiver DSP (Rx-DSP) are illustrated in Figure 7a,b.

We use the Friis free space propagation model to calculate the power budget of a wireless link [44,45,46]. Let d represent the distance between the transmitting antenna and the receiving antenna in free space. Assuming the transmitting antenna is an ideal omnidirectional antenna, the received signal power at the receiving point is calculated as follows:(13)$$P_{R}=\frac{P_{T}}{4\pi d^{2}}$$

In the formula, $P_{T}$ represents the transmitted power. Considering the directional gain $G_{T}$ of the transmitting antenna, the effective aperture $A_{R}$ of the receiving antenna, and the atmospheric loss $L_{\mathrm{am}}$, we obtain the following:(14)$$P_{R}=\frac{P_{T}}{4\pi d^{2}}G_{T}A_{R}L_{\mathrm{am}}$$

Converting the units to dBm and using the logarithmic model, we obtain the following:(15)$$P_{R}=P_{T}+G_{T}+G_{R}-L_{\mathrm{am}}-L_{\mathrm{FSP}}$$

The atmospheric loss in the terahertz band is approximately 10 dB/km. In a short-distance terahertz transmission system of 50 m, the impact of atmospheric loss is minimal. The free space path loss (FSPL) is as follows:(16)$$L_{\mathrm{FSP}}=20\mathrm{lg}\frac{4\pi d}{\lambda }$$

Table 2 outlines the power budget for each of the components used in our experimental system. When the input optical power of the UTC-PD is 14 dBm, the typical output power of UTC-PD at 300 GHz is roughly −8 dBm.

$G_{R}$ refers to the gain of the receiving antenna, including the lens and the horn feed (with a gain of 25 dB). Considering the coupling efficiency between the lens and the feed, imperfections in lens manufacturing precision, and errors caused by multipath effects in the indoor corridor, these losses are estimated to be around 25 dB. The calculated actual gain of the lens is approximately 45 dB, which is close to the theoretical calculation.

## 4. Experimental Validation

The experimental results of Figure 8a show a significant increase in bit error rate (BER) as the transmission rate increases. This indicates that, at higher transmission rates, the signal quality deteriorates, likely due to a reduced signal-to-noise ratio (SNR) caused by an increased bandwidth. At a transmission rate of 5 GBaud, the QPSK signal’s BER remains within the hard decision forward error correction (HD-FEC) threshold of 3.8 × 10^−3^. However, at 10 GBaud, although the BER exceeds the HD-FEC threshold, it still remains below the soft decision forward error correction (SD-FEC) threshold of 2.4 × 10^−2^. At 5 GBaud, the constellation points are well defined, indicating good signal integrity.

Figure 8b shows that BER significantly decreases as the input optical power increases. This trend suggests that increasing input power effectively improves signal quality and reduces BER. At 14 dBm input optical power, the signal’s BER meets the HD-FEC thresholds. The above results verify the practical application of the Fresnel lens we designed in terahertz systems.

Figure 8c illustrates the BER performance across different frequencies for 5 GBaud QPSK with 14 dBm input optical power. Since the lens is designed for 300 GHz, its efficiency drops significantly at other frequencies. The further away from the design frequency band, the worse the system performance becomes. This validates the frequency selectivity of the Fresnel lens. This means that the Fresnel lens performs best near the design frequency (295 GHz to 305 GHz), while its focusing ability and signal quality decrease significantly when operating away from the design frequency. This frequency selectivity characteristic makes the Fresnel lens particularly suitable for high-precision applications in specific frequency bands, but its performance may be limited in wideband applications.

## 5. Conclusions

In terahertz communication systems, transceiver antennas typically use plano-convex dielectric lenses, but these become bulky as the aperture size increases and the focal length decreases. To address this, we designed a Fresnel lens optimized for the terahertz band, balancing a large aperture and a short focal length. Through theoretical analysis and simulation, we evaluated its performance. A 300 GHz wireless communication system was built to test the Fresnel lens’ gain, and the results show it effectively amplifies signals, with a gain close to that of a standard plano-convex lens. The theoretical gain of a 30 cm Fresnel lens is 48.83 dB, while the actual measured gain is approximately 45 dB. In practice, due to factors such as installation errors, alignment errors, manufacturing precision, and aging, the actual gain may vary.

## Figures and Tables

**Figure 1 sensors-24-07592-f001:**
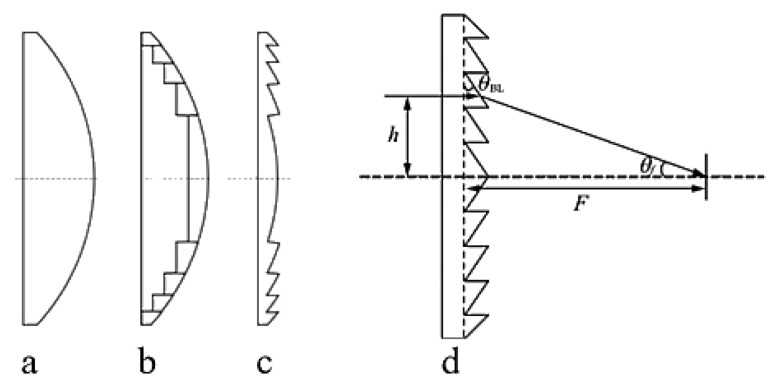
(**a**) Plano-convex lens; (**b**) curved-Fresnel lens; (**c**) planar-Fresnel lens; (**d**) planar-Fresnel lens design schematic.

**Figure 2 sensors-24-07592-f002:**
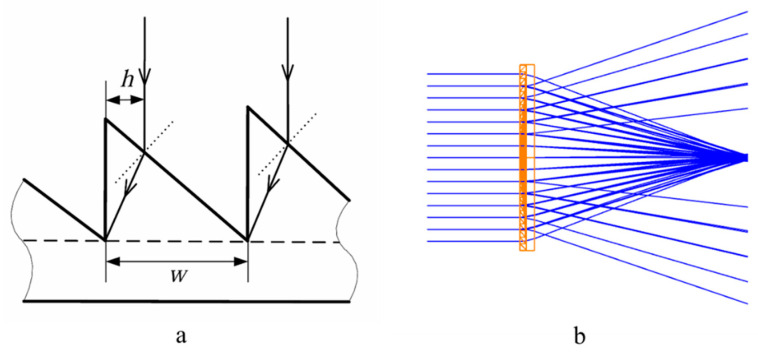
(**a**) Schematic diagram of the loss of lens structure. (**b**) The simulation diagram of off-axis scattering.

**Figure 3 sensors-24-07592-f003:**
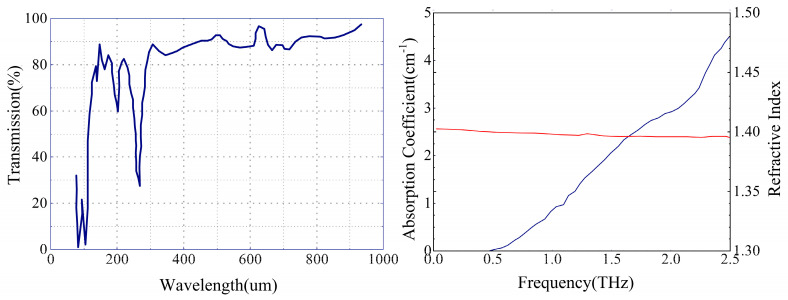
Transmission of PTFE film ~0.1 mm thick and THz refractive index and absorption coefficient in PTFE. The red line represents the refractive index curve, and the blue line represents the absorption coefficient curve.

**Figure 4 sensors-24-07592-f004:**
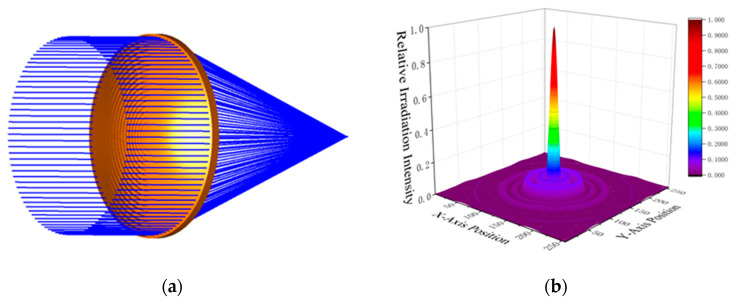
(**a**) Fresnel lens appearance and ray tracing; (**b**) Fresnel lens focal plane radiation intensity.

**Figure 5 sensors-24-07592-f005:**
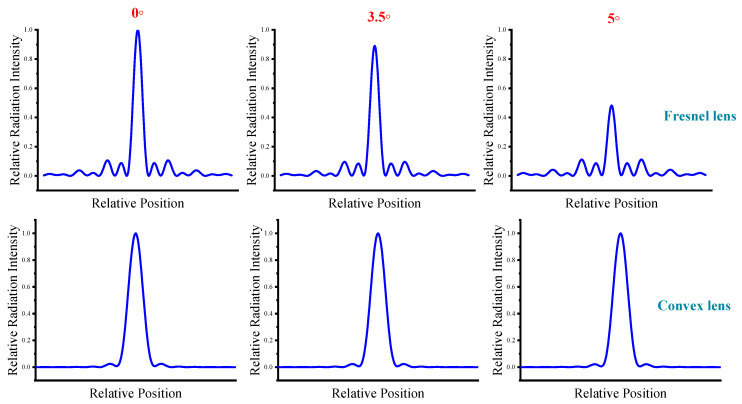
Effect of receiving angle on lens gain. The focusing ability of the Fresnel lens decreases as the incident angle increases, and the plano-convex lens is less affected.

**Figure 6 sensors-24-07592-f006:**
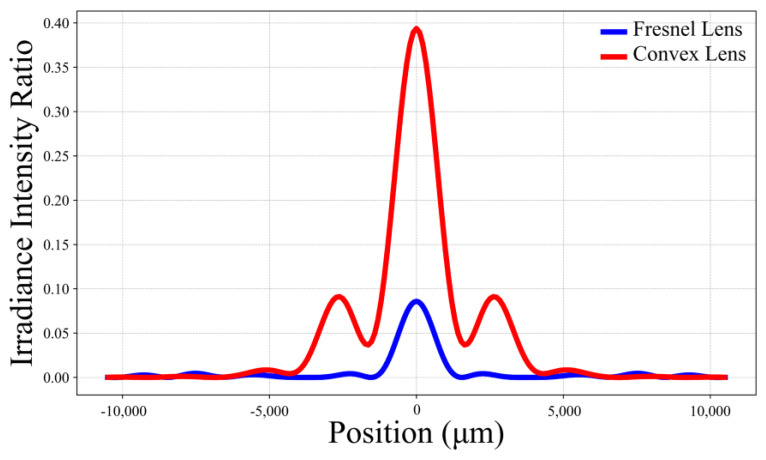
The relative irradiance intensity curves on the focal plane for the two types of lenses.

**Figure 7 sensors-24-07592-f007:**
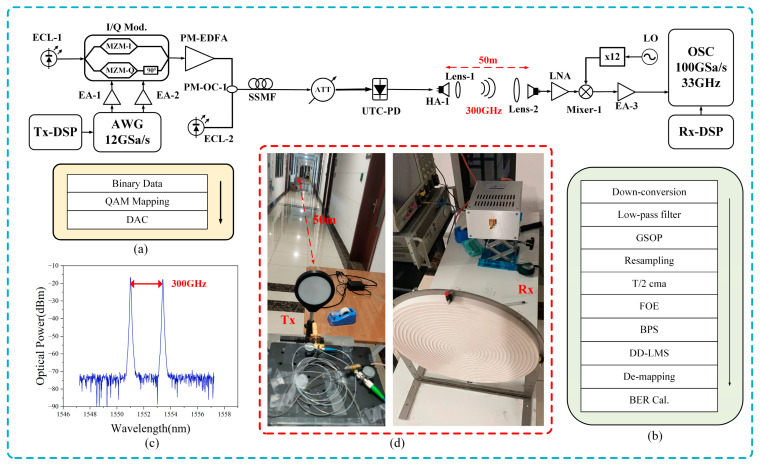
The experimental setup utilized for the wireless delivery of photonic Terahertz signals. (**a**) The digital signal processing (DSP) routine at the transmitter side. (**b**) The digital signal processing (DSP) routine at the receiver side. (**c**) The optical spectra of the input optical signals before UTC-PD. (**d**) Photograph of the experimental scene for 50 m wireless transmission.

**Figure 8 sensors-24-07592-f008:**
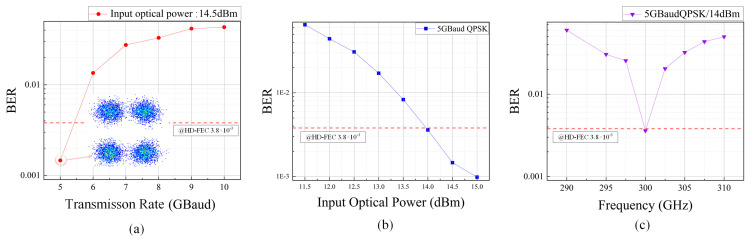
(**a**) BER vs. transmission rate at 14.5 dBm input optical power to UTC-PD. The constellation diagram is obtained by decoding the 5 GBaud signal. (**b**) BER vs. input optical power to UTC-PD for 5 GBaud QPSK signal. (**c**) BER vs. Frequency for 5 GBaud QPSK signal at 14 dBm input optical power to UTC-PD.

**Table 1 sensors-24-07592-t001:** Key equipment parameters in the experimental setup.

Parameter	Value
AWG	Sampling rate: 120 GSa/s
EA-1/EA-2	Gain: 25 dB
ECL-1	Wavelength: 1550.00 nm Linewidth: <100 KHz Output power: 12 dBm
ECL-2	Wavelength: 1552.4 nm Linewidth: <100 KHz Output power: 12 dBm
IQ Mod.	3 dB bandwidth: 30 GHz
UTC-PD	Frequency range: 280~380 GHz
HA-1/HA-2	Gain: 25 dB
Lens-1	Convex lens Diameter: 10 cm
Lens-2	Fresnel lens Diameter: 30 cm
LNA	Gain: 33 dB
Electrical LO	Frequency: 12.5 GHz
EA-3	Gain: 26 dB
OSC	Sampling rate: 100 GSa/s 3 dB bandwidth: 33 GHz

**Table 2 sensors-24-07592-t002:** Key equipment parameters in the experimental setup.

Parameter	Value
$P_{T}$	10 dB
$G_{T}$	40 dB
$L_{FSP}$	115.96 dB
$L_{am}$	0.5 dB
$G_{R}$	45 dB
$P_{R}$	−20 dB

## Data Availability

Data are contained within the article.

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
