# Peer review of "New Design Scheme for and Application of Fresnel Lens for Broadband Photonics Terahertz Communication"

_sensors, 2024, doi:10.3390/s24237592_

Round 1

Reviewer 1 Report

Comments and Suggestions for Authors

This manuscript can be considered for acceptance after minor revision, as the current version is not concise and refined enough.

additional comment 
I consider this work very good, so no much further revision required. For a communication system, the antennas always should be thin and light, so as the system easy to be implemented and used. Here the authors design a Fresnel lens for broadband Terahertz communication, not only simulations, but also experiments demonstrated the successful design. Though the Fresnel lens had been widely used for optical imaging and communication, while for Terahertz applications, still many chalenges have to face, including the materials, the fine structures and matching for communication applications. As here the manuscript clearly demonstrated the experimental results, which indicated that the lens work well for broadband Terahertz communications, I figure this work will push ahead the lightweight design and application of Fresnel lens antennas for broadband Terahertz communications.
Anyway, I think this work is good enough. But for the publishing, conciseness of the entire text and improvement of English grammar are necessary.

Comments on the Quality of English Language

The English expression is relatively clear, but not concise and refined enough.

Author Response

Thank you for your suggestion, I have made the corresponding changes

Reviewer 2 Report

Comments and Suggestions for Authors

The manuscript presents a terahertz Fresnel lens which balances the tradeoff between large aperture and short focal length. The authors carry out comprehensive theoretical design and communication-based experimental measurements, reaching solid results. Overall I would like to recommend an acceptance of the manuscript.  

Author Response

Thank you for your suggestion.

Reviewer 3 Report

Comments and Suggestions for Authors

I reviewed the manuscript titled "New Design Scheme and Application of Fresnel Lens for Broadband Photonics Terahertz Communication," in which the authors propose a Fresnel lens application for a long-distance Terahertz communication to overcome the limitations of traditional plano-convex lenses used as antennas in transceivers. The manuscript includes a detailed design, simulations, and experimental results that support the methods described. This work represents a valuable contribution to the research community as it addresses a significant research gap in long-distance Terahertz communication technology.

I recommend the manuscript for publication, subject to the following revisions:

(A) The primary objective of the paper is to design a Fresnel lens that achieves higher antenna gain with a larger diameter lens, which is impractical using a traditional plano-convex lens. However, the experiments were conducted with a smaller lens size, which could easily be built with a plano-convex lens. Could the authors justify why these experimental design parameters were chosen? Why not larger diameter Fresnel lens than 30 cm? Was it for cost-effectiveness or you wanted to compare with existing results? You project the similar results to the larger aperture lenses? 

(B) The authors designed and constructed a 30 cm Fresnel lens-based antenna, which was tested with a 300 GHz Terahertz signal over a 50-meter wireless link. They report a signal gain of 45 dB, which they compare to other works that use a 30 cm aperture-based plano-convex lens system over a separation of 850 meters, achieving a gain of 65 dBi.

Questions: (i) Please standardize the gain units (45 dB and 65 dBi) to allow for an accurate comparison. (ii) Could the authors explain the differences in gain between their experiment and the reference (Li et al., 2023)? In the manuscript, the transmitter and receiver are separated by 50 meters, while in Li et al. (2023), the separation was 850 meters. Given the shorter distance in your experiment, could the actual projected gain be less than 45 dB? Please clarify this in the text.

(C) I suggest revising the abstract to make it more concise and better aligned with the manuscript’s content. For example:

  • Line 10-11: "As the aperture increases and the focal length decreases, the thickness and mass of the plano-convex lens increase rapidly." This is not universally true. Are the authors referring to practical limitations in manufacturing large-aperture lenses? While a shorter focal length can allow for a wider field of view and increased signal gain, is this what you mean? This section could be clarified to avoid multiple interpretations.
  • Line 15-16: "The experimental results show that the Fresnel lens can amplify the signal in the terahertz band." Consider rewriting this as: "The experimental results demonstrate that the Fresnel lens can be used for long-distance Terahertz communication with larger aperture diameters, overcoming the limitations of traditional plano-convex lenses."
  •  

(D) The introduction is supported by appropriate and sufficient references, but the references should be cited throughout the text where relevant, rather than only at the end of the paragraphs. For example:

  • Line 1-4
  • Line 42-43
  • Line 70-71
  • Line 81-82 For example, [7] is appropriately used.

(E) Figures:

  • Figure 5: The caption needs more detail. Indicate that the angle of incidence decreases with flux. Additionally, please discuss why the Fresnel lens exhibits higher sidelobes compared to the plano-convex lens.
  • Figure 8: The figure caption is misplaced. It should be located directly below the figure, not after the "Experimental Validation" section heading.

(F) Add references for the following statement:

  • Line 56-57: "Typically, assuming structural losses are negligible, doubling the effective aperture of a plano-convex lens results in an approximate 6 dB increase in antenna gain." Please add a reference. Signal gain should increase with the square of the aperture size (D²).

Define polytetrafluoroethylene (PTFE) when it first appears at line 51, rather than later at line 275.

Reviewer 4 Report

Comments and Suggestions for Authors

This is a review of the manuscript titled “New Design Scheme and Application of Fresnel Lens for Broadband Photonics Terahertz Communication” by P. Tian and colleagues, submitted to the journal Sensors. The authors propose a design for a Fresnel lens operating at 300 GHz, utilizing both theoretical models and numerical simulations. The lens is fabricated from PTFE and experimentally validated. However, the manuscript has some shortcomings, particularly in providing sufficient details on the lens fabrication process and the comprehensiveness of the literature review. With significant revisions, the paper could be suitable for publication in Sensors.

I offer the following suggestions to improve the manuscript:

The literature review overlooks previous research on THz Fresnel lenses. A brief search revealed references [1] through [4], which should be acknowledged in the introduction.

It would also be useful to compare the performance of the proposed lens with other existing designs.

Although there are commercially available lenses [1], the manuscript’s novelty could be questioned. However, the fact that the current work includes data transmission distinguishes it from the existing literature.

How was the lens optimization performed? Did you use a brute-force search over the parameter space, or an optimization algorithm such as particle swarm optimization or gradient descent? Also, which parameter did you specifically optimize?

It may be beneficial to highlight the potential applications of Fresnel lenses beyond imaging.

In Figure 3, please specify which line represents the absorption coefficient and which one shows the refractive index.

For Figure 6, can you clarify the parameters of the convex and Fresnel lenses used in generating the plot? This will help provide context for comparing with the Fresnel lens.

In Figure 8, the inset of (a) needs to be clarified in the figure caption.

References:

[1] https://www.tydexoptics.com/products/thz_optics/thz-fresnel-lenses/

[2] https://opg.optica.org/view_article.cfm?pdfKey=c01c66a0-d928-48da-aa6518a8b73e623e_106939

[3] https://opg.optica.org/ol/fulltext.cfm?uri=ol-42-10-1875&id=363566

[4] https://iopscience.iop.org/article/10.1088/1674-1056/18/12/064/pdf

Comments on the Quality of English Language

English is good. Maybe check it for small grammar mistakes just in case.

Author Response

Thank you for your suggestions, they are of great help to us.

Lens optimization: We use the geometric optics method for optimization, the main optimization parameters are the cone coefficient, tooth width (frequency) and depth. The optimization algorithm is the damped least squares method.

The remaining suggestions have been modified in the corresponding positions.

Round 2

Reviewer 4 Report

Comments and Suggestions for Authors

My concerns were adequately addressed. However, I make the following minor recommendation. Please include a paragraph describing how you fabricated the lens (e.g. the machining process, etc.). I recommend the manuscript for publication.

Author Response

Thank you for your comments and suggestions. We have added the appropriate description in the first paragraph of section III:

After designing and planning the relevant parameters of the Fresnel lens, we generated the corresponding 3D model and manufactured it using a computer numerical control (CNC) machine.